# Bioactive Peptides Produced by Cyanobacteria of the Genus *Nostoc*: A Review

**DOI:** 10.3390/md17100561

**Published:** 2019-09-29

**Authors:** Anna Fidor, Robert Konkel, Hanna Mazur-Marzec

**Affiliations:** 1Division of Marine Biotechnology, Faculty of Oceanography and Geography, University of Gdańsk, Marszałka J. Piłsudskiego 46, PL-81378 Gdynia, Poland; anna.fidor@phdstud.ug.edu.pl (A.F.); robert.konkel@phdstud.ug.edu.pl (R.K.); 2Institute of Oceanology, Polish Academy of Sciences, Powstańców Warszawy 55, PL-81712 Sopot, Poland

**Keywords:** cyanobacteria, *Nostoc*, nonribosomal peptides, bioactivity

## Abstract

Cyanobacteria of the genus *Nostoc* are widespread in all kinds of habitats. They occur in a free-living state or in association with other organisms. Members of this genus belong to prolific producers of bioactive metabolites, some of which have been recognized as potential therapeutic agents. Of these, peptides and peptide-like structures show the most promising properties and are of a particular interest for both research laboratories and pharmaceutical companies. *Nostoc* is a sole source of some lead compounds such as cytotoxic cryptophycins, antiviral cyanovirin-N, or the antitoxic nostocyclopeptides. *Nostoc* also produces the same bioactive peptides as other cyanobacterial genera, but they frequently have some unique modifications in the structure. This includes hepatotoxic microcystins and potent proteases inhibitors such as cyanopeptolins, anabaenopeptins, and microginins. In this review, we described the most studied peptides produced by *Nostoc*, focusing especially on the structure, the activity, and a potential application of the compounds.

## 1. Introduction

Cyanobacteria, the photosynthetic Gram-negative bacteria, are one of the oldest forms of life on Earth. As oxygen producers, they made a tremendous impact on the evolution of life on our planet. Today, their role as abundant primary producers is also important [1]. The species of the N_2_-fixing *Nostoc* genus (order Nostocales) belong to the most common cyanobacteria. They occur in terrestrial ecosystems as well as in fresh, brackish, and marine waters, living in a free form or as symbionts in association with marine sponges [2], cycads [3], or as a component of cyanolichens [4]. *Nostoc* has a wide geographical distribution, and it was reported from different parts of the world—from Arctic and Antarctic to tropical regions [5,6,7,8,9]. The well-developed adaptive strategies enable the cyanobacterium to withstand repeated desiccation, extreme temperatures, salt stress, UV-radiation, and pathogen infections [10,11,12,13]. Due to a high tolerance to extreme conditions, *Nostoc* was considered to be a good candidate for extraterrestrial agriculture [14].

For hundreds of years, the cyanobacteria of the genus *Nostoc* have been used as herbs and/or healthy food for people. High contents of fiber, amino acids, proteins, vitamins, and carbohydrates increase their nutritional value. *Nostoc*-containing food products are still consumed in China, India, Indonesia, Peru, Ecuador, and Bolivia [15,16,17,18]. *Nostoc* has also been applied as biofertilizer [19,20] and a rich source of bioactive compounds, including anticancer [21], antifungal [22], antibacterial [23], antiviral [24,25,26], and enzyme inhibiting [27] agents. These metabolites were identified as peptides, lipopeptides, fatty acids, alkaloids, and terpenoids [28,29,30,31,32]. The pharmaceutical application of peptides and peptide-like structures has been explored most widely; the compounds frequently represent a promising starting point for the design of novel drugs [33]. Peptides tend to bind selectively to cellular targets, reducing the risk of side effects. The drug-like properties of small peptides, such as plasma half-life, bioavailability, and selectivity, can be improved. In addition, some peptides have more than one function, and the combined effects of their activities can be observed, e.g., antimicrobial and immune system stimulating activity [34,35].

This review focuses on the most widely studied peptides produced by cyanobacteria of the genus *Nostoc*. The structure and the biological activity of the biomolecules are described. Both the cyanobacterium and its metabolites represent a high potential for biotechnological application.

## 2. Non-Ribosomal Peptides (NRPs) and Polyketides (PKs)

A significant part of the metabolites produced by *Nostoc* belongs to nonribosomal peptides (NRPs) or polyketides (PKs). They are characterized by a structural diversity and a broad spectrum of biological activities, including cytotoxic [36], enzyme inhibiting [37], anti-inflamatory [38], antibacterial [39], and antifungal effects [40]. NRPs are mainly synthesized by bacteria (Proteobacteria, Actinobacteria, Firmicutes, and Cyanobacteria) and fungi (Ascomycota) [41]. The ability to produce NRPs and PKs is a strain-specific feature, and in an individual strain, several classes of the compounds can be found. These compounds have linear, cyclic, or branched cyclic structure and are composed of proteogenic and non-proteogenic amino acids. They also contain short fatty acid chains, amines, heterocyclic rings such as thiazole or oxazole, and other residues [32]. NRPs, PKs, and their hybrids (NRPs/PKs) are synthesized on multifunctional enzyme complexes with a modular structure called non-ribosomal peptide synthetases (NRPS) or polyketide synthases (PKS) [28]. Each module of the complex is subdivided into domains catalyzing specific reactions in the multi-step process of residue incorporation into the peptide chain. In NRPS, the adenylation domain selects and activates a specific amino acid residue. Then, the residue is linked to the peptidyl carrier protein via thiol-containing phosphopantetheine. The condensation domain in the final module releases the peptide from the NRPS and terminates the chain elongation [28,42,43,44]. The structural modifications of NRPs are introduced by tailoring enzymes catalyzing methylation, oxygenation, cyclization, halogenation, glycosylation, and epimerization reactions. The cyanobacterial PKs are assembled on type I PKS with modules consisting of at least three domains: the acyltransferase domain, which recognizes and activates the substrate, the acyl carrier protein, which transports the molecule on the active site of ketosynthase involved in a bond formation [28], and the thioesterase domain which releases the final product [28,42]. Similarly to NRPS, the synthesized PKs are modified by tailoring enzymes, such as ketoreductase, dehydratase, enoylreductase, or methyl transferase. The modifications often lead to increase in the bioactivity and the resistance of NRPs and PKs to enzymatic lysis [45]. They also generate large structural diversity within one class of compounds.

*Nostoc* is one of the most prolific producers of NRPs/PKs (Appendix A) [46]. Some classes of cyanopeptides are common to several cyanobacterial genera (e.g., microcystins, anabaenopeptins, cyanopeptolins), while others have been identified only in species belonging to the genus *Nostoc* (e.g., nostocyclopeptides, cryptophycins, nostopeptolides). In this work, we focus mainly on the peptides produced solely by the genus *Nostoc*, but some examples of the peptides occurring in other cyanobacterial taxa are discussed as well.

## 3. Cryptophycins

For the first time, cryptophycins [Crs, molecular weight (MW) 604–688 Da], the 16-membered cyclic depsipeptides, were isolated from the lichen associated *Nostoc* sp. ATCC 53789 (Arron Island, Scotland) [47]. The name of this class of peptides is related to their potent antifungal activity against the yeast of the genus *Cryptococcus* [48]. Crs were also found in *Nostoc* sp. GSV-224 (ATCC55483) from a terrestrial sample collected in India [36], in *Nostoc* sp. ASN_M from paddy fields in Iran [30], and in the Okinawan marine sponge *Dysidea arenaria* [2]. The Cr identified in the sponge, arenastatin A, turned out to be identical to Cr-24 from *Nostoc* sp. ATCC 53789. This finding raised the question about the real origin of the compound. More than 28 natural Crs were identified in *Nostoc*, and many structural variants were synthesized [36,49]. In cyanobacteria, the compounds are assembled on a mixed PKS/NRPS enzyme complex [50]. Their biosynthesis is encoded in two PKS genes (*crp*A and *crp*B), two NRPS genes (*crp*C and *crp*D), and tailoring genes (*crp*E-H) encoding enzymes involved in epoxidation and chlorination reactions. Interestingly, the Cr biosynthetic gene clusters in the two Cr-producing cyanobacteria, ATCC53789 and GSV-224, are identical [50].

The natural Crs consist of four building blocks—ABCD (amino or hydroxycarboxilic acids)—linked in a cyclic sequence through an ester bond between C and D (Figure 1) [51]. Cr-1 belongs to the most abundant natural variants. It contains phenyloctenoic acid (A), 3-chloro-O-methyl-d-tyrosine (B), methyl-β-alanine (C), and L-leucic acid (D). Crs showed strong antiproliferative and cytotoxic effects against both solid and hematologic tumor cell lines, including multidrug-resistant (MDR) cancer cell lines [52,53]. Crs isolated from *Nostoc* sp. GSV 224 were found to be active against human nasopharyngeal carcinoma (KB), human colorectal adenocarcinoma (LoVo) and human ovarian carcinoma (SKOV3) cells in pM to nM range [36]. Crs also showed activity against a murine leukemia (L1210) cell line [54]. 

Crs bind to tubulin protein, causing inhibition of microtubule assembly as well as the suppression of microtubule dynamics [54]. These effects lead to cell cycle arrest in the G2/M phase and cell death through apoptosis. Rapid uptake of Crs and accumulation in cells results in a prolonged activity. The compounds are poor substrates for P-glycoprotein (P-gp), the membrane transporter that mediates excretion of xenobiotics out of cells. P-gp are overexpressed in cancer cells, which contributes to the development of MDR [54]. The optimal cytotoxic activity is achieved in the case of cryptophycins with intact 16-membered macrolide structure, reactive epoxide ring in unit A, methyl group in units A (C-6) and C (C-2), *O*-methyl group and chloro-substituent in unit B, and isobutyl group in unit D [36,49]. Modification introduced to the structure by conversion of the epoxide group in unit A into chlorohydrin significantly increased the cytotoxicity of the compounds, but in the aqueous solution, the obtained derivative was found to be unstable [55,56]. This problem was overcome by the synthesis of stable glycinate ester analogues with retained activity of cryptophycin chlorohydrin [53,57]. Cr-52 (LY355703), a synthetic analogue of Cr-1 with an additional methyl group in unit C, was widely explored as a promising anticancer agent. It reached phase II clinical trials for non-small cell lung cancer and in patients with platinum-resistant advanced ovarian cancer. Unfortunately, due to unacceptable toxicity and lack of efficacy in vivo, the tests on Cr-52 were discontinued [58,59]. However, the work on Cr as anticancer drug has not been completely ceased. In recent years, the introduction of innovative, highly targeted methods opened new possibilities for drug development. The efficacy of cryptophycins can be improved by conjugation through a cleavable linker with an antibody or small molecule. The non-toxic antibody-drug conjugates (ADCs) or small molecule-drug conjugates (SMDCs) have improved stability in plasma and selectively target cancer cells. In tumor cells, the toxic peptide is released and then initiates the processes of cell death [60,61].

## 4. Nostocyclopeptides

Nostocyclopeptides (Ncps; MW 756–881 Da) are a small class of nonribosomal heptapeptides. The first two Ncp variants that differ only in one residue, Ncp-A1 (Tyr**^1^**-Gly**^2^**-Gln**^3^**-Ile**^4^**-Ser**^5^**-MePro**^6^**-Leu**^7^**) (Figure 1) and Ncp-A2 (Tyr**^1^**-Gly**^2^**-Gln**^3^**-Ile**^4^**-Ser**^5^**-MePro**^6^**-Phe**^7^**), were isolated from a terrestrial strain *Nostoc* sp. ATCC 53789 from India [62]. Nostocyclopeptide M1 (Ncp-M1) (Tyr**^1^**-Tyr**^2^**-HSer**^3^**-Pro**^4^**-Val**^5^**-MePro**^6^**-Tyr**^7^**) was obtained from *Nostoc* sp. XSPORK 13A isolated from a gastropod collected at the Cape of Porkkala (Baltic Sea) [63]. Ncps are characterized by a unique imino linkage formed between the amine group of the conserved l-Tyr in position 1 and the aldehyde hydrate of the residue in position 7. The head-to-tail intramolecular cyclization reaction is selective and reversible in aqueous solution [64]. In the known Ncps, the C-terminal position is most variable (Leu, Phe, or Tyr), while position 6 is always occupied by a methylated form of Pro (MePro) (Figure 1). The residues in position 3 are d-amino acids [62,63]. The Ncp biosynthetic gene cluster described by Becker et al. [65] contains two large NRPS genes: the *ncp*A gene encoding a three-module protein NcpA1-3 and the *ncp*B gene encoding a four-module protein NcpB1-4. The co-linear arrangement of the gene cluster, the organization of the modules, as well as a substrate specificity of the adenylation domains determine the structure of Ncps. For example, the NcpA3 module contains an epimerase domain, which corresponds to the presence of d-amino acid in position 3, while the terminal part of the NcpB protein contains a reductase domain corresponding to the presence of aldehyde group in the C-terminus of the linear form of Ncp [65].

According to Golakoti et al. [62], Ncp-A1 and Ncp-A2 showed a weak cytotoxic activity (IC_50_ ca. 1 µM) against a human nasopharyngeal cancer cell line (KB) and a human colorectal cancer adenocarcinoma cell line (LoVo). On the other hand, Ncp-M1 was nontoxic against primary rat hepatocytes [63]. In these cells exposed to microcystin-LR (MC-LR), nostocyclopeptide Ncp-M1 abolished apoptotic effects of the hepatotoxin. It was found that the three nostocyclopeptides as well as the synthetic analogue of Ncp-M1 with all L-amino acid residues and without imino bond blocked the uptake of MC-LR and nodularin (NOD) to rat hepatocytes and to the human embryonic kidney cells (HEK293). The antitoxin activity of Ncps was attributed to the inhibition of the organic ions transporters OATP1B3 and OATP1B1, which facilitate the uptake of the toxins into the cell [63,66]. Of the tested compounds, Ncp-M1 counteracted the MC-LR-induced apoptosis in the most potent way.

## 5. Cyanovirin-N

Cyanovirin-N (CV-N; MW 11 kDa) is one of the most widely studied cyanobacterial lectins, with antiviral activity at µM to pM level. CV-N was isolated from *Nostoc ellipsosporum* maintained in the culture collection at the University of Hawaii [24]. This carbohydrate-binding polypeptide consists of 101 amino acids, including four cysteine residues that form two intra-chain disulfide bonds (Cys8-Cys22; Cys58-Cys73) (Figure 1). These bonds stabilize the CV-N structure and determine the antiviral activity of the polypeptide [67]. The molecule represents a unique structure with an extremely low sequence homology to any other known proteins.

CV-N is organized in two domains characterized by a high sequence duplication (32%) and structure identity. Domain A contains residues 1–38 and 90–101, and domain B contains residues 39–89 (Figure 1) [67,68]. In each domain, there are two binding sites for carbohydrates separated by the ~40 Å distance and with 10-fold difference in affinity [69]. In solution, the natural CV-N is mainly present as a monomer, while in crystals, dimmer is formed by a strand exchange across the two domains [25,68]. Proline in position 51 plays an important role in the swapping of the domains and monomer aggregation. CV-N is a stable compound and preserves its activity even after treatment with denaturants, detergents, organic solvents, or extremely high temperatures (100 °C, 15 min) [24]. Conversion of dimmer into monomer is extremely slow, but at higher temperature (>38 °C), the process is accelerated [70].

CV-N showed a potent in-vitro and in-vivo activity against human immunodeficiency virus (HIV-1 and 2), simian immunodeficiency virus (SIV), and other enveloped viruses [24,26,71,72]. It specifically and strongly interacted with the viral envelop glycoprotein gp120 through *N*-linked high mannose glycans (Man-8 and Man-9). Consequently, the gp120 could not bind to the host CD4 T-cell receptor and the chemokine CCR5 and CXCR4 co-receptors. Thus, the viral entry as well as the cell-to-cell fusion and transmission were inhibited [24,26,69,73,74].

Due to good stability, lack of toxic effects (as a 0.5–2% gel), broad spectrum of antiviral activities, and highly specific binding to oligosaccharides, CV-N offers a great potential for prevention of HIV sexual transmission. The female macaques (*Macaca fascicularis*) treated with a CV-N gel as a topical vaginal microbicide were found to be resistant to a pathogenic chimeric SIV/HIV-1 virus, SHIV89.6P [75]. Positive effects and a 63% reduction in the transmission of the virus were also obtained in macaques dosed vaginally with *Lactobacillus jensenii* 1153–1666 expressing CV-N [76,77]. Similar results were reported for CV-N as a preventive measure in rectal transmission of SHIV89.6P in macaques [75,78]. In the macaques models, CV-N was proven to be non-toxic and had no other adverse effects.

In order to obtain CV-N in the amounts required for further studies and at low costs, this polypeptide was expressed in several bacterial hosts (e.g., *Escherichia coli*, *Lactococcus lactis*, *Lactobacillus plantarum*) as well as in yeast and transgenic plants [79]. Cytoplasmic expression of CV-N in *E. coli* gave 3–4 mg of pure and bioactive recombinant protein (rCVN) per g of wet biomass [80,81]. In case of *Nicotiana tabacum*-biosynthesized rCVN, the yield was 130 mg of monomeric form per g fresh leaf tissue [82]. Products with increased activity were obtained by expression of fusion proteins containing CV-N and the plant-derived HIV-neutralizing monoclonal antibody mAb b12 [83] or the *Pseudomonas* exotoxin PE38 [84]. To reduce the immunogenic effects and the risk of proteolysis and to increase the half-life of the molecule in serum, a covalent binding of CV-N to poly(ethylene glycol) (PEGylate) was suggested [85]. As the PEGylation sites were at or near the mannose binding sites of CV-N, in many cases, loss of activity was observed. The in vitro activity was preserved only in the case of the PEGylated mutant Q62C with glutamine 62 replaced by cysteine and with the extra free sulfhydryl group. More successful results were obtained for PEGylated linker-extended CV-N with (Gly4Ser)3 at the N-terminus designed by Chen et al. [86].

## 6. Other Peptides Exclusively Produced by *Nostoc*

Cyanobacteria of the genus *Nostoc* produce numerous other peptides, including nostophycin, nostosin, nostopeptolides, nostoweipeptins, and banyasin (Appendix A). Some of the peptides have been identified only in cyanobacteria of this genus or even only in one *Nostoc* strain. Therefore, the available data about their structure and activity are scarce and sometimes limited to one to three papers.

Nostophycin (MW 888 Da, Figure 2), the cyclic hexapeptide with a novel β-amino acid residue, was detected only in *Nostoc* sp. 152 isolated from the lake Sääksjärvi in Finland [87]. The general structure of the compound is Ahoa[Pro-Ile-Phe-Pro-Gly-Gln] where Ahoa represents the unique β-amino acid residue, 3-amino-2,5-dihydroxy-8-phenyloctanoic acid (Figure 2). This unit is present only in nostophycin. Other examples of β-amino acids in cyanopeptides include 3-amino-9-methoxy-2,6,8-trimethyl-10-phenyl-4,6-decadienoic acid (ADDA) in microcystins [88], 3-amino-2-methylhexanoic acid (Amha) in medusamide [89], and 2-methyl-3-aminopentanoic acid (Map) in urumamide [90]. Nostophycin is synthesized constitutively on a hybrid PKS/NRPS enzyme complex encoded by the *npn* gene cluster [91]. The relationship between the net production rate of the compound and *Nostoc* sp. 152 growth was observed. However, under physiological stress induced by phosphorous and light limitation, the cell content of nostophycin increased up to 10-fold [92]. Nostophycin was found to be non-toxic (at 20 µg/mL) against bacteria (*Staphylococcus aureus*, *Bacillus subtilis*, *Escherichia coli*) and fungi (*Aspergillus niger*, *Candida albicans*) and showed weak cytotoxicity (10 µg/mL) against the lymphocytic mouse leukemia L1210 cells [87].

The cyclic depsipeptides, nostopeptolides (Np; MW 1033–1080 Da, Figure 2) belong to the MePro-containing peptides. They were found in cryptophycin-producing *Nostoc* sp. GSV224 [93], *Nostoc* sp. UK2almI isolated from lichen [94], and *Nostoc punctiforme* PCC73102 [95]. The general structure of the three Nps (A1–A3) from GSV224 is BA-IleA1/ValA2)[Ser-MePro-LeuAc-Leu-Gly-Asn-Tyr-Pro], where MePro and leucylacetate (LeuAc) are the two non-proteinogenic amino acids and BA is a butyryl group. Np A3 was suggested to be either an epimer of Np A1 or an artifact generated during sample processing [93]. The four Nps (L1–L4) from *Nostoc* sp. UK2almI have the following sequence of residues, MHEA[Thr-*O*-MeSer-PhePr-MeProL1,L2/ProL3,L4-Dhb-Ile-Gln-HypL1,L3/ProL2,L4], where MHEA is 2-methylhex-2-enoic acid, *O*-MeSer is *O*-methylated serine, PhePr is phenylalanylpropanoic acid, Dhb is dehydrobutyric acid, and Hyp is hydroxyproline (Figure 2) [94]. As it could be concluded from the presence of non-proteinogenic amino acid residues in the structures, Nps belong to NRPs and are assembled through a mixed NRPS/PKS biosynthetic pathway [96]. Screening of 116 cyanobacterial genomes for the presence of genes involved in the biosynthesis of MePro (nosE and nosF) showed that, in contrast to other cyanobacterial genera, these genes are quite common in *Nostoc* [94].

Besides nostopeptolides, MePro is also present in the structure of nostocyclopeptides and nostoweipeptins (W; MW 1172–1214 Da, Figure 2). The latter class of peptides was found in *Nostoc* sp. XPORK 5A isolated from the Baltic Sea at Porkkala (Gulf of Finland). In four of the seven identified nostoweipeptins, two MePro residues were present. The structure of the major nostoweipeptin W1 is *N*-MePhe-Ac[Ser-MePro-Leu-Ile-Tyr-Ser-Hyp-Hyp-MePro]; an ester bond is located between MePro and Ser (Figure 2). Similarly as in the case of nostocyclopeptides, nostopeptolides and nostoweipeptins block the transport of microcystin and nodularin to hepatocytes HEK293 through OATP1B1/B3. The inhibition of the hepatotoxin-induced apoptosis by nostopeptolides and nostoweipeptins at µM concentrations indicates a possible application of the peptides as potent antitoxins [94].

Nostosins (Ns-A; MW 449 Da and Ns-B; MW 451 Da, Figure 2), the linear nonribosomal tripeptides, were detected in *Nostoc* sp. FSN from a paddy field in Iran [97]. The peptides are composed of 2-hydroxy-4-(4-hydroxyphenyl)butanoic acid (Hhpba) in N terminus, Ile in the second position, and argininal (Ns-A) or argininol (Ns-B) in C terminus (Figure 2). In the cell extract from *Nostoc* sp. FSN, minor amounts of four other nostosin variants (Ns-C-F) were also detected. In these compounds, Ile was replaced by Val (Ns-D and F), and Hhpba was replaced by deoxyHhpba (Ns-C and E). Nostosins belong to serine proteases inhibitors. Ns-A with argininal is more potent and inhibits the activity of porcine trypsin with IC_50_ of 0.35 μM, while the IC_50_ value of Ns-B is 0.55 μM [97]. The strong trypsin-inhibiting activity of Ns-A is in line with similar effects observed for other small linear cyanopeptides with terminal argininal, e.g., spumigin E [98] or aeruginosin [99].

## 7. Peptides Produced by *Nostoc* and Other Cyanobacteria

As many other cyanobacteria, *Nostoc* produces bioactive nonribosomal peptides such as microcystins, cyanopeptolins, microginins, and anabaenopeptins. There are also reports on the presence of the rybosomaly synthesized microviridins in *Nostoc minutum* NIES-26 [100] and *N. punctiform*e PCC73102 [95] (Appendix A). The structure, the activity, and the biosynthetic pathways of the peptides have been described in many scientific papers (e.g., [28,32,101,102]). Here, we refer only to some distinctive features of the peptides produced by cyanobacteria of the genus *Nostoc*.

Microcystins (MCs, Figure 3), the hepatotoxic cyclic heptapeptides, belong to the most widely studied cyanobacterial metabolites. In cyanobacteria of the genus *Nostoc*, microcystin variants with O-acetyl-demethyl Adda (ADMAdda) instead of Adda (Figure 3) are common and almost exclusively present in the members of this genus (Figure 3) [103,104,105,106,107]. The interactions of microcystins with protein phosphatases (PP1 and PP2a) and their toxicity strongly depend on a cyclic structure and the Adda-Glu part of the molecules [108]. In mouse bioassay, the ADMAdda-containing MCs (Figure 3) exhibited similar toxicity as the most potent variants of this class of peptides (LD_50_ of 100–200 µg/kg) [104]. *Nostoc* is also the rare example of non-*Nodularia* producer of nodularin, the cyclic pentapeptide with a structure similar to microcystins [109,110]. Cyanopeptolins (CPs, Figure 3) constitute another class of peptides commonly detected in different taxonomic groups of cyanobacteria. They are composed of a six-amino acid ring with a unique 3-amino-6-hydroxy-2-piperidone (Ahp) residue (Figure 3). The side chain is linked to the cyclic part via the amino group on Thr [111]. In *Nostoc*, the compounds with similar structures are known as nostopeptins [37,112], insulapeptolides [113], and nostocyclins [114]. Nostopeptins A and B (BA^A^/Ac^B^-Gln[Hmp-Leu-Ahp-Ile-MeTyr-Ile]) and insulapeptolides (variant D: Ac-Cit[Hmp+Leu+Ahp+Ile+diMeTyr+Ile]) differ from CPs produced by other cyanobacteria in the presence of 3-hydroxy-4-methylproline (Hmp) instead of Thr [37,112,113]. A distinct feature of nostocyclin ([Thr+Hse+Ahp+Phe+MeTyr+Val]Hse+Ile+Hpla) is the presence of hydroxyphenyllactic acid (Hpla) in the side chain and two homoserine residues (Hse)—one in a side chain and one in the ring part of the molecule [114]. The CPs with the structure typical of this class of compounds (Figure 3) identified in other cyanobacteria (e.g., *Microcystis* or *Planktothrix*) were detected in *Nostoc* sp. TAU strain IL-235 from the spring pool of the Banyas stream in Israel [37] and *Nostoc edaphicum* CCNP1411 from the Baltic Sea [27]. Cyanopeptolin-like compounds produced by *Nostoc* inhibited activity of serine proteases (trypsin, chymotrypsin, or/and elastase) at nM to low µM concentrations [37,112,113]. Nostocyclin was active against protein phosphatases PP1 but at a relatively high IC_50_ value 64 µM [114].

Anabaenopeptins (Aps) were detected in different cyanobacterial genera, including *Nostoc*, strains ASN-M [30], PCC73102 [115,116], and CENA543 [117]. The peptides are composed of a five-amino acid ring with a side chain attached through an ureido linkage (Figure 3). In *Nostoc* sp. CENA543 isolated from a Brazilian saline lake, six APs were found, including four new variants named nostamide with a general structure Ile/Val-CO[Lys-Ile/Val-Hph-MeAla/Ala-Hph/Phe], where Hph is homophenylalanine and MeAla is methylalanine (Figure 3). The strain also produces four namalides with anabaenopeptin-like scaffold but lacking two residues in the ring part (Ile/Val-CO[Lys-Ile/Val-Hph/Hty] [117]. This type of peptide was earlier reported from marine sponge *Siliquariaspongia mirabilis* collected off Nama Island (Federated States of Micronesia) [118] and from the Brazilian strain of cyanobacterium *Sphaerospermopsis torques-reginae* ITEP-024 [119]. Anabaenopeptins usually inhibit activity of carboxypeptidases [120,121], protein phosphatases [122], or/and serine proteases (trypsin, chymotrypsin) [27,37]. Namalides were found to be active against carboxypeptidase A at submicromolar to micromolar concentrations [117,119].

Nostoginins, the structural analogues of microginins, are another example of the differences within the same class of peptides produced by *Nostoc* and other cyanobacteria. In the C-terminal position of nostoginins, 3-amino-2-hydroxy decanoic acid (Ahda) residue present in microginins is replaced by 3-amino-2-hydroxy octanoic acid (Ahoa) (e.g., Ahoa-Val-MeIle-MeTyr) (Figure 3) [37]. Nostoginin BN741 (MW 741 Da) isolated by Ploutno and Carmeli (2002) inhibited the activity of bovine amino peptidase with IC_90_ of 1.3 M.

## 8. Conclusions

A high number of peptides produced by cyanobacteria of the genus *Nostoc*, their unique structures, and a wide spectrum of biological activities indicate a significant biotechnological potential of the organism. In case of already known bioactive peptides of potential pharmaceutical application, efforts are made to improve their drug-like properties by structure modification or conjugation with antibodies or small molecules. To produce sufficient amounts of bioactive agents for preclinical and clinical studies, the effectiveness of chemical synthesis and cloning is explored. In parallel, the search for new peptides produced by *Nostoc* is continued. Surprisingly, despite a wide geographical distribution and common occurrence of *Nostoc*, the most studied metabolites of the genus were detected in a limited number of strains. As new cyanobacterial culture collections with *Nostoc* isolates have recently been established in different institutions, and the use of methods for peptides detection, isolation, and activity screening are more common, further progress in the drug development based on the metabolites produced by *Nostoc* can be expected.

## Figures and Tables

**Figure 1 marinedrugs-17-00561-f001:**
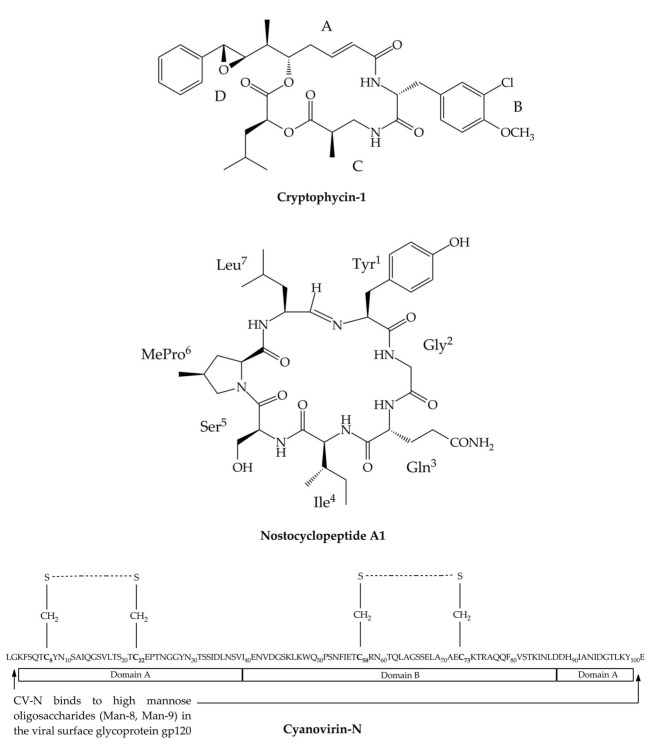
Structure of peptides produced by the genus *Nostoc*: Cryptophycin 1 (Cr-1), nostocyclopeptide A1 (Ncp-A1), and cyanovirin-N (CV-N). In Cr-A, the four building blocks (ABCD) represent phenyloctenoic acid (A), 3-chloro-*O*-methyl-d-tyrosine (B), methyl-β-alanine (C), and l-leucic acid (D).

**Figure 2 marinedrugs-17-00561-f002:**
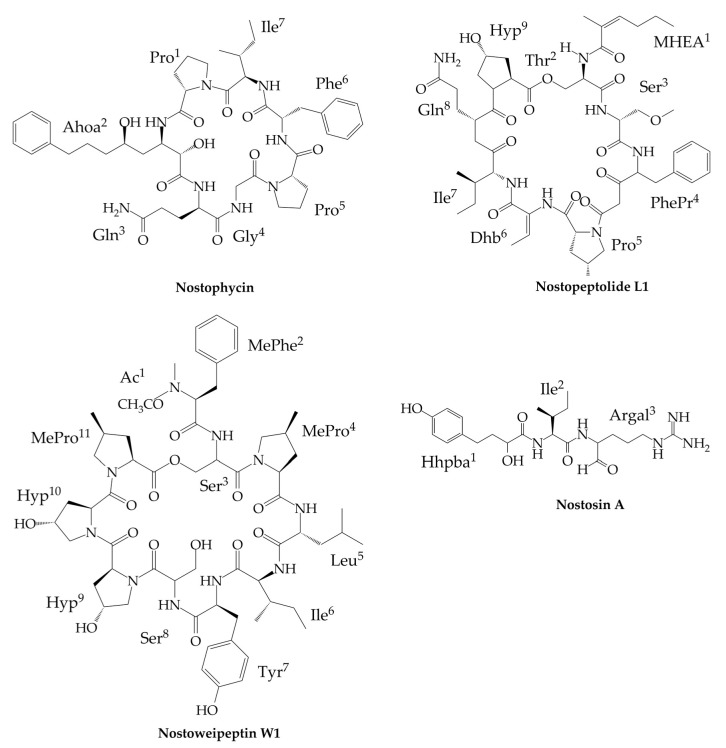
Structures of nostophycin, nostopeptolide L1 (Np-L1), nostoweipeptin W1 (Nwp-W1), and nostosin A (Ns-A) produced exclusively by cyanobacteria of genus *Nostoc*.

**Figure 3 marinedrugs-17-00561-f003:**
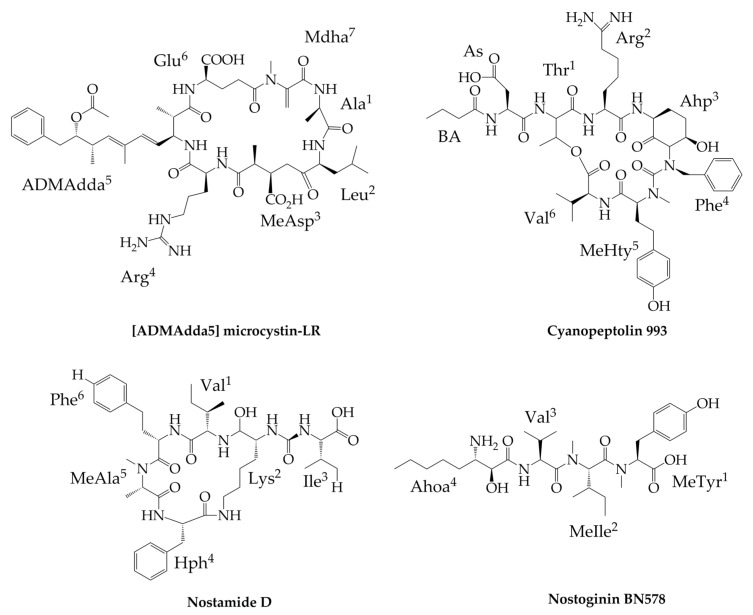
Structures of [ADMAdda5] microcystin-LR ([ADMAdda5]MC-LR), cyanopeptolin 993 (CP993), nostamide D (Na-D), and nostoginin BN578 produced by cyanobacteria of genus *Nostoc* and other cyanobacteria.

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
