# Peer review of "Bioactive Peptides Produced by Cyanobacteria of the Genus *Nostoc*: A Review"

_marinedrugs, 2019, doi:10.3390/md17100561_

Round 1
Reviewer 1 Report
A review of bioactive peptides produced by cyanobacteria of the
genus Nostoc was discussed in detail.
A lot of peptides are displayed in the supplementary materials.
This review should be published.
They conclude that "A high number of peptides produced by cyanobacteria of the genus Nostoc, their unique structures and a wide spectrum of biological activities, indicate significant biotechnological potential of the organism."
Future work on the cyanobacteria of the genius Nostoc should be mentioned in the conclusion.
Comparison to similar structures should also be mentioned in the supplementary materials to attract readers attention.
Author Response
Rev. 1:…..This review should be published.
They conclude that "A high number of peptides produced by cyanobacteria of the genus Nostoc, their unique structures and a wide spectrum of biological activities, indicate significant biotechnological potential of the organism."
Future work on the cyanobacteria of the genius Nostoc should be mentioned in the conclusion.
Answer: General comments on the main challenges of further research on Nostoc bioactive peptides are added to Conclusions.
Rev. 1: Comparison to similar structures should also be mentioned in the supplementary materials to attract readers attention.
Answer: This comment is unclear to us. The peptides produce by Nostoc, and their corresponding structures produced by other cyanobacteria, are described in the manuscript. In the revised version of our work, the Supplementary materials include only the list of peptides identified in Nostoc.
Reviewer 2 Report
This paper works on the bioactive (not all) peptides from the Nostoc cyanobacteria. Certainly cyanobacteria are widely recognized as very prolific sources of natural peptides and peptide-bearing compounds. The genus Nostoc is doubtlessly among the most frequently encountering cyanobacteria for natural products research. Therefore, a peer review on Nostoc-derived peptides would be quite informative to natural products community.
Given these, this review tried to cover major groups of Nostoc-derived peptides. Firstly, these compounds were categorized into three subgroups: three extensively studied ones (sections 3-5), less-studied ones from exclusively Nostoc genus (section 6), and those from Nostoc along with other cyanobacteria (section 7). Then these compounds were step-wisely reviewed on their structural features, biosynthesis, bioactivities, and other activities. The writing is concise and easy to read. Overall this review paper would be acceptable at Marine Drugs.
The authors are urged to reconsider and/or revise the following flaws:
Unlike the three groups of representative ones, the drawings of less-studied peptides from only Nostoc (section 6), and those from Nostoc and other cyanobacteria (section 7) are not included in main text. A representative compound from each structural group in these sections should be added in main text (or move from SI).
All of the bioactivities of these peptides should be reported by their IC50 or similar numerical values to provide objective information to readers.
Supporting Information: In nostocyclopeptides section, what does nostocyclopeptide (MW 774) stand for? The reference (ref 33) of this compound is not a research paper but a recent review. If this compound is a real one and isolated prior to nostocyclopeptides A1-A3, the section 4 should be re-written including the information on this compound.
Minors,
Lines 56-57: “antiinflamatory” should read “anti-inflammatory” Also “antimicrobial” and “antifungal” should combine together to “antimicrobial” or re-sort to “antibacterial” and “antifungal” Line 133: The seventh residue of Ncp-A1 must be not “Ile7” but “Leu7” (see Fig. 1). Line 226: The section title would change to contain “exclusively” for peptides produced only by Nostoc. Line 245: MePro unit is not very rare in natural peptides. Indeed this unit is also found in nostocyclopeptides and others (section 4 and lines 260-261). Therefore this sentence should be re-written. It is seemed that compounds in SI were arranged by their alphabetical orders. Those lacking these, such as cyanopeptilins and microcystins should be orderly rearranged by either their molecular weights, strain numbers or uncommon amino acid residues.
Author Response
Rev 2: …. Therefore, a peer review on Nostoc-derived peptides would be quite informative to natural products community. ….The writing is concise and easy to read. Overall this review paper would be acceptable at Marine Drugs. The authors are urged to reconsider and/or revise the following flaws:
Unlike the three groups of representative ones, the drawings of less-studied peptides from only Nostoc (section 6), and those from Nostoc and other cyanobacteria (section 7) are not included in main text. A representative compound from each structural group in these sections should be added in main text (or move from SI).
Answer: The peptide structures have been moved from SI to the main text. So now, the structures of all classes of the discussed peptides are shown in the work.
Rev. 2. All of the bioactivities of these peptides should be reported by their IC50 or similar numerical values to provide objective information to readers.
Answer: Some data on the activity was already included in the previous version of the work. When the data refer to a class of peptides, a level of activity was indicated (e.g. low mM). In the current version of the manuscript, the lacking IC50 values were added, if available.
Rev. 2: Supporting Information: In nostocyclopeptides section, what does nostocyclopeptide (MW 774) stand for? The reference (ref 33) of this compound is not a research paper but a recent review. If this compound is a real one and isolated prior to nostocyclopeptides A1-A3, the section 4 should be re-written including the information on this compound.
Answer: We wish to thank the reviewer for this comment. We use the incorrect data published by other authors. This mistake is corrected in the current version of SI.
Rev. 2: Lines 56-57: “antiinflamatory” should read “anti-inflammatory” Also “antimicrobial” and “antifungal” should combine together to “antimicrobial” or re-sort to “antibacterial” and “antifungal”
Anwer: Corrected and more relevant reference was added ([39])
Rev. 2: Line 133: The seventh residue of Ncp-A1 must be not “Ile7” but “Leu7” (see Fig. 1).
Answer: Corrected
Rev. 2: Line 226: The section title would change to contain “exclusively” for peptides produced only by Nostoc.
Answer: Thank you for valuable comment. Corrected.
Rev. 2: Line 245: MePro unit is not very rare in natural peptides. Indeed this unit is also found in nostocyclopeptides and others (section 4 and lines 260-261). Therefore this sentence should be re-written.
Answer: We agree. Nostoc produces several classes of peptides with MePro. The word “rarely” was deleted. We just meant that MePro is a rare component of peptides, in general.
Rev. 2: It is seemed that compounds in SI were arranged by their alphabetical orders. Those lacking these, such as cyanopeptilins and microcystins should be orderly rearranged by either their molecular weights, strain numbers or uncommon amino acid residues.
Answer: Corrected.